# Tactile sensitivity and motor coordination in infancy: Effect of age, prior surgery, anaesthesia & critical illness

Laura Cornelissen[1,2]*, Ellen Underwood[1,2¤a], Laurel J. Gabard-Durnam[3], Melissa Soto[1,2¤b], Alice Tao[1,2¤c], Kimberly Lobo[1,2], Takao K. Hensch[2,4], Charles B. Berde[1,2]

1 Department of Anesthesiology, Critical Care & Pain Medicine, Boston Children's Hospital, Boston, MA, United States of America, 2 Harvard Medical School, Boston, Massachusetts, United States of America, 3 Center for Cognitive and Brain Health, Northeastern University, Boston, Massachusetts, United States of America, 4 F.M. Kirby Neurobiology Center, Department of Neurology, Boston Children's Hospital, Boston, MA, United States of America

¤a Current address: New York University Grossman School of Medicine, New York, NY, United States of America
¤b Current address: Johns Hopkins University, Baltimore, Maryland, United States of America
¤c Current address: Weill Cornell Medical College, New York, NY, United States of America
* laura.cornelissen@childrens.harvard.edu

**Data Availability Statement:** All relevant data are within the paper and its Supporting Information files.

## Abstract

### Background

Tactile sensitivity in the infant period is poorly characterized, particularly among children with prior surgery, anaesthesia or critical illness. The study aims were to investigate tactile sensitivity of the foot and the associated coordination of lower limb motor movement in typically developing infants with and without prior hospital experience, and to develop feasible bedside sensory testing protocols.

### Materials and methods

A prospective, longitudinal study in 69 infants at 2 and 4 months-old, with and without prior hospital admission. Mechanical stimuli were applied to the foot at graded innocuous and noxious intensities. Primary outcome measures were tactile and nociceptive threshold (lowest force required to evoke any leg movement, or brisk leg withdrawal, respectively), and specific motor flexion threshold (ankle-, knee-, hip-flexion). Secondary analysis investigated (i) single vs multiple trials reliability, and (ii) the effect of age and prior surgery, anaesthesia, or critical illness on mechanical threshold.

### Results

Magnitude of evoked motor activity increased with stimulus intensity. Single trials had excellent reliability for knee and hip flexion at age 1-3m and 4-7m (ICC range: 0.8 to 0.98, p >0.05). Nociceptive threshold varied as a function of age. Tactile sensitivity was independent of age, number of surgeries, general anaesthesia and ICU stay.

**Funding:** This work was supported in part by the Boston Children's Hospital Anaesthesia Trailblazer Research Award (LC); Sara Page Mayo Endowment for Pediatric Pain Research, Education, and Treatment (CBB); and World Premier International Research Center Initiative– International Research Center for Neurointelligence (TKH; https://www.jsps.go.jp/english/). Funding sources had no role in study design, data collection and analysis, decision to publish, or preparation of the manuscript.

**Competing interests:** The authors have declared that no competing interests exist.

## Conclusions

This brief sensory testing protocol may reliably measure tactile and nociceptive reactivity in human infants. Age predicts nociceptive threshold which likely reflects ongoing maturation of spinal and supraspinal circuits. Prior hospital experience has a negligible global effect on sensory processing demonstrating the resilience of the CNS in adverse environments.

## Introduction

Early life experience plays an important role in shaping neurodevelopment across multiple sensory domains. Tactile and pain processing are important for everyday function from the first few days of life onwards. Being held skin-to-skin by a caregiver immediately post-birth is a key mediator of early maternal bonding. Skin-to-skin contact regulates an infant's pain-related behaviour, physiologic and cortical response following an acute noxious (skin-breaking) procedure [1, 2]. Deprivation of touch stimulation during childhood is associated with physical and cognitive deficits [3]. While maturation of tactile and pain processing are well characterised in adults and older children [4–6], little has been described about how tactile sensitivity develops in infancy.

Lower limb flexion responses inform how tactile information is processed and conveyed between the skin, spinal cord, and brain [4, 7]. In adults, the magnitude of the flexion response is correlated with the amount of perceived pain [8]. Human newborn infants are very sensitive to cutaneous mechanical stimulation and display prolonged and exaggerated lower limb flexion responses to tactile and noxious stimulation of the foot, which gradually decrease with age [9–11].

Designing sensory testing protocols in infants is challenging—infants are unable to follow instructions or provide verbal report of sensation, and can easily sensitise or habituate to repeated stimulation [12, 13]. Recent efforts have been made to limit the burden of laboratory research protocols including testing feasibility of single trials in adults [14, 15]. There is a critical need to identify brief and feasible sensory testing protocols that can be performed in pre- and non-verbal populations and provide useful comparative data for research and clinical purposes. The results of these quantitative sensory tests have implications for use in research and in the clinic when examining nerve function for pain assessment [16], behavioural disorders [17], and in clinical trials of analgesics and local anaesthetics [18, 19], and in safety testing of novel spinal therapeutics for neurodegenerative disorders [20].

Hospitalised infants are necessarily exposed to a sensory environment that is dramatically different from a home environment. Hospital admissions commonly consist of clinically required procedures that are considered painful or distressing (e.g. surgery, venepuncture, intubation), and often require treatments that rely heavily on touch (i.e. handling, repositioning) [21]. The maturation of touch sensitivity and pain processing in infancy is thought to be shaped following early adverse experiences in the first few months of life—a highly sensitive period of sensory development. There is concern that repeated early life stressors, both painful and non-painful, during sensitive windows of development produce somatosensory memory that exerts a long-lasting influence on later sensory processing and associated pain behaviour [22–24]. Yet, we understand little of how factors associated with early-life hospital experience (multiple surgeries, prolonged general anaesthesia, and critical illness) interact to shape the development of the somatosensory system.

## Study aims

Studies of tactile and nociceptive responses in paediatrics have focused primarily on premature and term infants [11, 12, 25, 26], or on older children of the age and cognitive development that requires the use of verbal responses [5, 6, 25, 27, 28], and generally provide data from a single point in time.

To address the gap in the literature, we performed a longitudinal study to examine tactile and nociceptive processing in a large group of infants at 1 to 3 months and at 4 to 7 months old. These infants were born at- or near-term, and were comprised of three groups according to prior hospital admission and general anaesthesia (GA) exposure: Inpatient (no history of GA exposure), Inpatient (GA-exposed), and Controls (i.e., no previous hospital admission and no-GA).

The first aim of this study was to examine motor responses to graded stimulation and the associated mechanical sensory thresholds in the skin during early infancy. We hypothesised that greater stimulus intensities generated a larger motor response, and that tactile sensitivity and nociceptive reactivity are influenced by postnatal age.

The second aim was to evaluate the trial agreement of sensory thresholds obtained using a single trial and multiple trials. This aim was driven by the need to develop brief quantitative sensory testing protocols that can be both performed at the bedside and easily applied in clinical trials, particularly in populations who cannot provide self-report. We hypothesised that single trials are feasible and yield suitable data for sensory assessments in human infants.

Our final aim was to analyse whether infant sensory thresholds are related with early-life hospital admission, surgery, or critical illness. We hypothesised that tactile and nociceptive thresholds are influenced by factors associated with prior hospital experience.

# Materials and methods

## Subjects

Children in the present study were part of a prospective, longitudinal investigation across the first 5 postnatal years of infants who had multiple, single and no exposure to general anaesthesia in the first few months of life. The main investigation is currently ongoing with anticipated completion in December 2025. Institutional review board approval was obtained from Boston Children's Hospital (IRB-P00028129) prior to starting the study. The study was conducted at Boston Children's Hospital. Written, informed consent was obtained from parents or legal guardians prior to their child's participation in the study.

Sixty-nine children were recruited from the medical and surgical wards at Boston Children's Hospital and from the community between July 2018 and March 2020. Inclusion criteria were age 1 to 7 months old (when uncorrected for prematurity). Exclusion criteria were gestational age at birth < 32 weeks-old, congenital malformations or other genetic conditions associated with sensory or motor impairment, characteristics that prevent free limb movement (i.e. leg cast for hip dysplasia). No children were acutely unwell, or mechanically ventilated at the time of study. **Fig 1** shows the flow of participants through the study.

Each child was invited to participate in the study on two occasions according to their age. First, at age 1 to 3 months old, and again at age 4 to 7 months old. Children took part in sensitivity testing to assess tactile sensory function. Parents or legal guardians completed a questionnaire concerning their child's demographics and clinical history. At the end of the study visit, parents/guardians were given a $25 gift card as an honorarium. Parents/guardians received an additional $5, $10 or $15 gift card based on the distance travelled, to assist with travel costs.

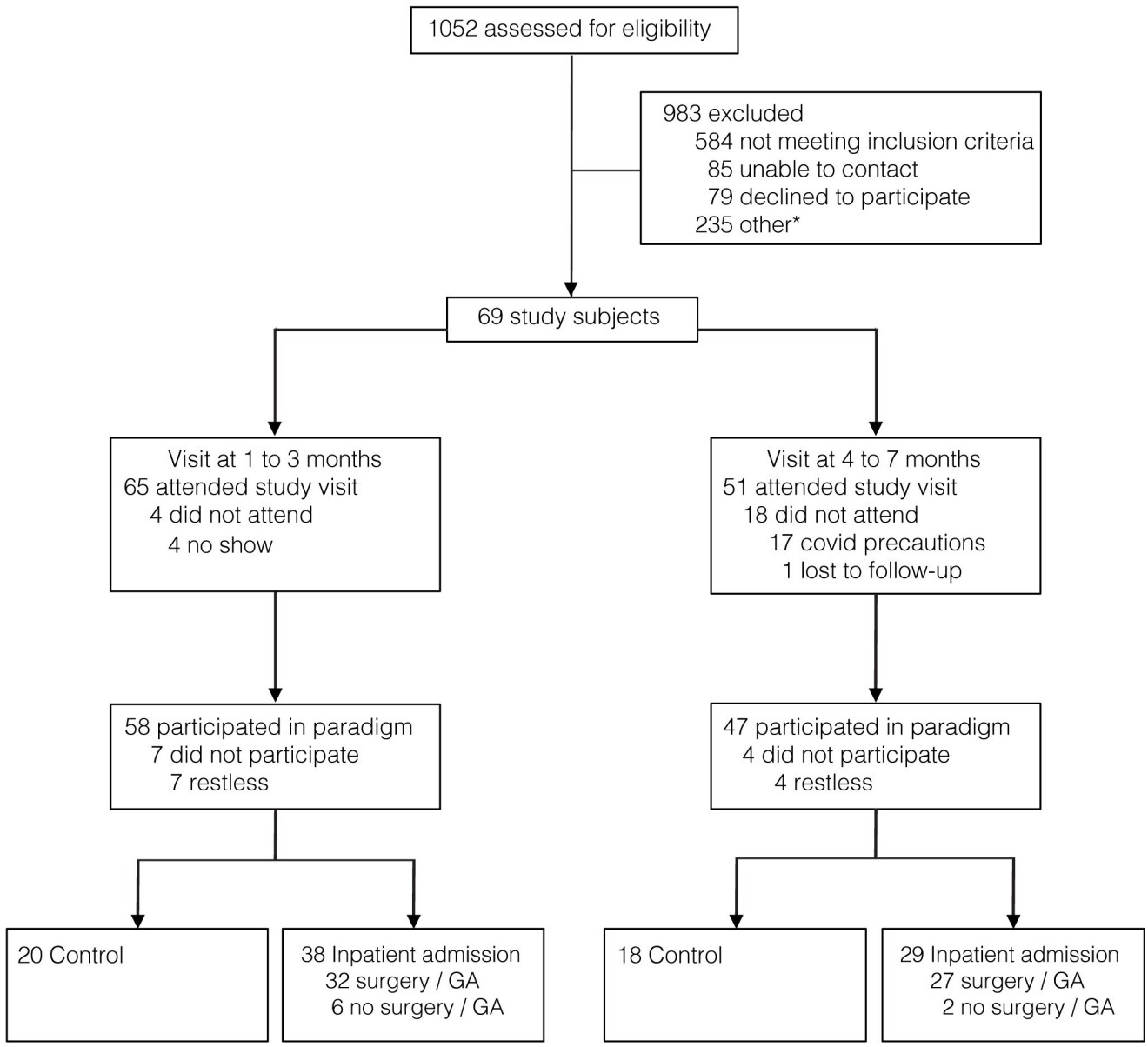

**Fig 1. Flow of participants through the study.** * 'Other' indicates children who did not meet stratified recruitment targets at the time of screening.

## Sensory testing protocol

Tactile sensitivity was evaluated using von Frey's monofilaments (North Coast Medical, CA). The Von Frey monofilament test is a classical measure of sensitivity to tactile pressure that is used for diagnostic and research purposes. In this test, the tip of a fibre with a specific weight (from 0.008 to 300g) is pressed against the skin at right angles, bending for 1 second on a contact area of 0.5mm in diameter. The force of the application increases as the researcher advances the probe until the fibre bends.

In this study, each participant was tested in a quiet room or at the hospital bedside. Participants were seated in an upright position on the lap of their caregiver with feet dangling (1–3 months: 40 out of 66, 62%; 4–7 months: 41 out of 49, 84%) or lying down. Single stimuli at

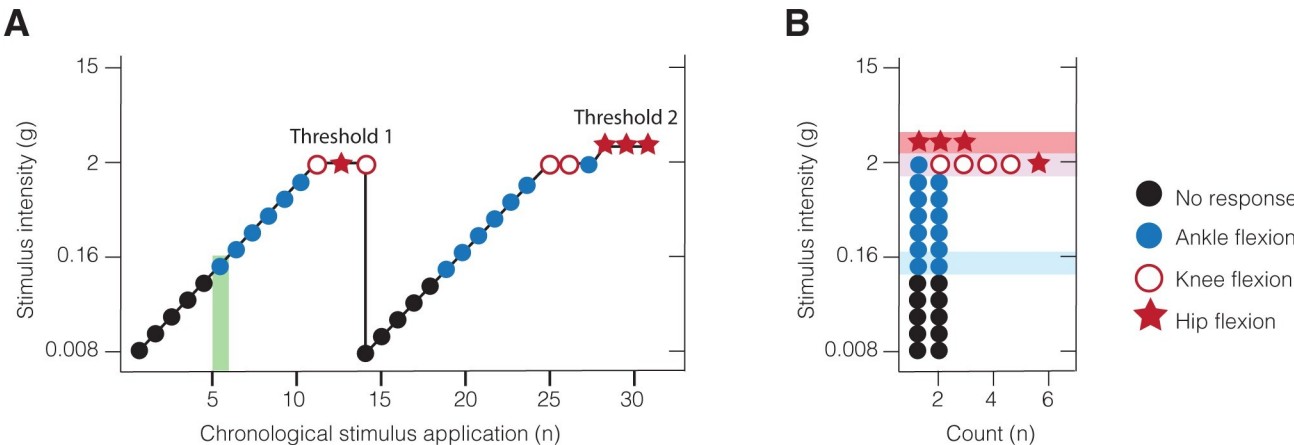

**Fig 2. Study design.** (A) Schematic of Tactile Sensitivity paradigm in an example subject. Tactile sensitivity threshold, identified as the force required to evoke the first response, is shown in green shaded area (i.e. stimulus # 6 using a force of 0.16g). Nociceptive threshold is marked as "Threshold 1" and "Threshold 2". (B) Frequency distribution of motor-responses in an example subject. The lowest stimulus intensity required to generate reproducible (a) ankle-flexion, (b) knee-flexion or (c) hip-flexion using data from all trials is determined (shaded areas show in corresponding colour as appropriate). Black circle represents no response, blue circle represents ankle-flexion, pink circle represents knee flexion, and red star represents hip-flexion.

graded weights were applied to the plantar surface of the foot (at the level of the 1<sup>st</sup> to 3<sup>rd</sup> metatarsus head) at rest. Gross motor activity was recorded following stimulus application. Gross motor activity consisted of either (i) ankle-flexion (dorsiflexion), (ii) knee-flexion, and (iii) hip-flexion. The stimuli were applied in ascending order, starting with the lightest force (0.008g). The trial threshold was established at the force when the brisk leg withdrawal consisting of knee or hip flexion was observed 3 times with the same fibre. The trial was then repeated. Testing was complete if the two trial threshold values were within 2 hair-units (see **S1 Table in S1 File** for conversion from hair-unit to absolute value in force-grams). If the difference in hair-unit required to evoke brisk leg withdrawal was greater than 2-hair-units, then a third trial was performed. **Fig 2** shows the study design.

Video recordings (iPhone 5S, Apple inc., CA) captured the subject's motor response following each stimulus application for post-hoc analysis. The rear-facing camera on the smartphone was used to capture 1080p HD video at 30 frames per second. Demographics and clinical characteristic data were collected from electronic health records or from the parent/legal guardian questionnaire.

## Data analysis

**Gross motor-response assessment.** Gross motor-response was assessed post-hoc using video recordings of the study visit. The assessor was a researcher who was blinded the participant's medical and clinical history. Motor activity was inspected frame-by-frame (i.e. one frame per 30ms) (VLC media player, France). Gross motor response was assigned to one of four categories that were defined by the how the lower leg flexed: (1) no-response (leg remains at rest); (2) ankle-flexion (brisk movement of the foot away from the stimulus, bending movement is limited to the ankle); (3) knee-flexion (brisk removal of leg away from stimulus, bending movement is limited to the lower leg- thigh remains in contact with surface); or (4) hip-flexion (brisk removal of leg away from stimulus, movement brings knee and thigh upward toward the chest).

**Outcome measures.** The primary outcome measures were Tactile Sensitivity threshold and Nociceptive Threshold expressed in grams. Tactile Sensitivity threshold was the lowest

stimulus intensity required to generate any visible motion of the ankle, knee or hip. Nociceptive Threshold was the average stimulus intensity required to evoke brisk leg withdrawal away from the stimulus, consisting of knee- or hip-flexion. Secondary outcome measures were (i) ankle-flexion (dorsiflexion), (ii) knee-flexion, and (iii) hip-flexion threshold, expressed in grams.

## Statistical analysis

Analyses were conducted using R, version 4.1.2 [29].

**Data processing.**   Sample data were split into three groups according to prior hospital admission and exposure to GA: (1) Inpatient (no-GA); (2) Inpatient GA-exposed; and (3) Control group (i.e., no previous hospital admission and no-GA). Sample data were then assigned to one of two age groups according to corrected age at the time of study: 1 to 3 months-old, and 4 to 7 months-old. Premature born children were defined as birth before 37 weeks of gestation. Preterm born children were age corrected by subtracting the number of weeks born before 40 weeks of gestation from the chronological age [30].

Data were tested for normality of distribution by Shapiro-Wilk tests [31]. Threshold variables were logarithmically transformed before analysis to achieve a near-normal distribution. Log-transformed data were used for the calculation of t-tests, and mixed-model analysis. P values $< 0.05$ were considered statistically significant. Absolute values in grams are reported in tables and used in figures, unless otherwise stated. Percentage, median with interquartile range, and mean with standard deviation are provided.

**Missing data and interrater reliability of motor scoring.**   Missing data occurred in individuals whom stimulus intensities greater than the threshold force (i.e. where brisk leg withdrawal consisting of knee- or hip flexion is evoked) were not applied. We opted to limit stimulus application to the threshold force in this infant sample for ethical and practical purposes. Missing values were replaced using the last observation carried forward approach (see **S1 Table in S1 File**).

Inter-rater agreement of motor response from video assessment was determined between one pair of observers to obtain the weighted Cohen's kappa. Motor response was re-scored by two researchers (EU, LC) in random subset of videos (18% sample; 505 stimulus applications and motor responses). Videos were epoched to short segments to ensure the assessors were blinded to the stimulus intensity. Inter-rater scores showed substantial agreement with 78% observations in agreement, and 26% observations in agreement expected by chance (weighted Cohen's kappa was 0.76).

**Single vs multiple trials for determining threshold.**   The agreement between measurements when using a single trial (T1) and "average of all trials" (TAvg) were evaluated in the Control group. Trial thresholds were compared for three types of lower limb flexion responses (ankle, knee, hip) in two age groups (1 to 3 months, and 4 to 7 months). Agreement between measurements was evaluated using intraclass correlation coefficients (ICC), paired t-tests and Bland-Altman analysis following reporting guideline recommendations on QST reliability studies [32].

Intraclass correlation coefficients (ICC, two-way mixed effects, multiple measurements (3, 2) model) were computed to quantify the consistency of the two measurements [33]. ICC values $> 0.81$ were considered "excellent"; between 0.61 and 0.8 were "good"; between 0.41 and 0.60 were "moderate"; between 0.21 and 0.4 were "acceptable", and $\leq 0.2$ were "poor" reliability [34]. ICC analyses were performed with the *irr* R package [35].

Differences between measurements were tested for significance using paired t-tests that control for the variability between subjects. P-values were adjusted to avoid Type-I error

(false-positive findings) due to multiplicity using Bonferroni correction [36]. Bland-Altman plots were used to depict the difference between measurements [37]. Limits of Agreement were calculated by the formula: mean observed difference ±1.96 x standard deviation of observed differences. If Limits of Agreement were small and the mean of the differences between measurements was near 0, then the tests were considered agreeable. Bland-Altman analyses were performed with the *blandr* R package [38].

**Examination of risk factors on threshold.** Linear mixed model analyses were performed to assess whether infant age and prior hospital admission (i.e., surgery, anaesthesia or critical illness) predicted sensory threshold. This type of modelling allows the relationship between covariates and reported responses with time-varying predictors (e.g. age) to be analysed, random effects that quantify variation, and handling of missing data without the need for explicit imputation (i.e. missed study visit) [39]. Linear mixed model analyses were performed with the *lme4* R package [40].

Tactile sensitivity threshold and nociceptive threshold were the dependent variable in separate models. Fixed effects were examined in a series of models: Model 1 assessed whether age (in months) and number of surgeries at the time of study predicted sensory threshold. Model 2 assessed whether age (in months) and cumulative duration of general anaesthesia (in hours) at the time of study predicted sensory threshold. Model 3 included age (in months) and cumulative duration of general anaesthesia (in hours), with ICU stay (in days) as a measure of critical illness. An additional model incorporated age, sex, and state, factors known to affect sensory processing and lower limb reflex withdrawal properties in adults [4, 41]; (**S2 Table in S1 File**).

All models included subjects as a random factor to control for variation from subject to subject. Intraclass Correlation Coefficients (ICC) were computed by dividing the between-group variance by the total variance i.e., sum of between-group and within-group (residual) variance. ICC values close to zero indicate low correlation between variance measures. Within-group variance ($\sigma^2$) and between-group-variance (variation between individual intercepts and average intercept, $\tau_{00}$) were provided following recommendations [42].

Collinearity of the variables was tested using the Variance Inflation Factor—a measure of the magnitude of collinearity between one variable and the remaining explanatory variables. Variance Inflation Factor scores were considered to be a 'low' correlation when $< 5$; all variables in each model had scores $< 2$ and were acceptable. Collinearity of the variables was tested using the *performance* R package [43].

Models were compared to select the best approximating model based on Akaike Information Criterion (AIC) [44]. AIC considers how well candidate models fit the dataset relative to each of the other models by using likelihood estimation and the number of fitted parameters. Sample-size adjusted AIC (AIC$_c$) scores were ranked (the smaller the value, the better the model). The relative strength of each candidate model was compared to the top ranked model based on the difference in AIC$_c$ values ($\Delta_i$), Akaike weight and Evidence Ratio. Criterion for accepting a model were (1) $\Delta_i < 2$ AIC units (indicating the model is as good as the best approximating model); and (2) cumulative Akaike weight $< 0.95$ [45]. Evidence Ratio provides a measure of how much more likely the best model is than other models. Models 1, 2 and 3 had $\Delta_i < 2$ AIC units, fell within cumulative Akaike weights of 0.87 (indicating 87% confidence that one of the remaining models is the best approximating model), and were considered acceptable for predicting tactile and nociceptive threshold. Model 4 did not satisfy AIC criteria. Model selection was tested using the *AICcmodavg* R package [46].

**Power calculation.** The current study is a secondary analysis based on data from a larger ongoing study examining the effect of early life general anaesthesia exposure on child development.

**Table 1. Demographics and clinical characteristics of the sample separated by age group and inpatient history at the time of study.**

| n | 1 to 3m | | | 4 to 7m | | |
|---|---|---|---|---|---|---|
| | Control | Inpatient Hx No Surgery/GA | Inpatient Hx + Surgery/GA | Control | Inpatient Hx No Surgery/GA | Inpatient Hx + Surgery / GA |
| | **21** | **7** | **38** | **19** | **2** | **28** |
| Age at birth, wks | 38.65 (1.28) | 37.35 (2.88) | 37.05 (2.25) | 38.57 (1.41) | 37.57 (3.63) | 38.24 (1.74) |
| Age at study, m | 2.31 (0.48) | 2.06 (0.83) | 2.46 (0.90) | 4.78 (0.38) | 4.25 (0.35) | 4.80 (0.34) |
| Sex, male, n (%) | 11 (52) | 2 (29) | 31 (82) | 13 (68) | 0 (0) | 21 (75) |
| **Surgical history** | | | | | | |
| Surgery, n (%) | | | | | | |
| 0 | 21 (100) | 7 (100) | 0 (0) | 19 (100.0) | 2 (100.0) | 0 (0.0) |
| 1 | 0 (0) | 0 (0) | 31 (82) | 0 (0.0) | 0 (0.0) | 22 (79) |
| $\geq 2$ | 0 (0) | 0 (0) | 7 (18.4) | 0 (0.0) | 0 (0.0) | 6 (21.4) |
| Age at 1st surgery, days | - | - | 27.00 [2.00, 44.50] | - | - | 33.50 [1.75, 64.00] |
| Duration of GA, hrs | - | - | 3.77 [2.05, 6.92] | - | - | 5.08 [1.78, 7.89] |
| **Admission history** | | | | | | |
| Inpatient stay, n (%) | - | 7 (100)[a] | 30 (79) | - | 2 (100.0)[b] | 25 (89) |
| Age at 1st stay, days | - | 0 [0.00, 12.00] | 0.00 [0.00, 34.50] | - | 11.00 [5.50, 16.50] | 0.00 [0.00, 28.00] |
| Duration of stays, days | - | 3 [2.00, 10.00] | 17.00 [11.75, 32.50] | - | 4.00 [3.50, 4.05] | 14.00 [7.00, 26.00] |
| ICU stay, n (%) | - | 6 (86)[c] | 26 (68) | - | 2 (100)[d] | 22 (79) |
| Age at 1st stay, days | - | 0.5 [0.00, 1.75] | 0.00 [0.00,21.25] | - | 11.50 [6.25, 16.75] | 0.00 [0.00, 0.75] |
| Duration of stays, days | - | 3.50 [2.00, 12.50] | 9.00 [5.00, 16.00] | - | 3.50 [2.75, 4.25] | 6.00 [5.00, 9.75] |

Data are given as mean (SD), or median [LQ, UQ], or n with % unless otherwise stated. Duration of stays or surgeries are taken as cumulative duration between birth and study visit. Abbreviations: GA = general anaesthesia; ICU = intensive care unit; L/UQ = lower/upper quartile; m = months; SD = standard deviation, wks = weeks.

[a.] 6 participants required an ICU stay; 1 participant required propanol treatment of haemangioma for 1 day.

[b.] 2 participants required an ICU stay.

[c.] 1 participant required feeding support for 15 days; 1 participant required monitoring following a brief, resolved unexplained event (i.e. when an infant stops breathing, has a change in muscle tone, turns pale or blue in colour, or is unresponsive) for 5 days; 1 participant had prenatal diagnosis of bladder exstrophy and required monitoring for 4 days; 1 participant required glucose monitoring for 2 days; 1 participant required oxygen level monitoring due to fluid in lungs following precipitous delivery for 2 days; 1 participant required monitoring due to diagnoses of laryngomalacia.

[d.] 2 participants required neonatal monitoring due to diagnoses of laryngomalacia and bladder exstrophy respectively.

## Results

### Study participants

Table 1 shows the characteristics of the 69 infants who were recruited. Fifty-eight infants were studied when they were aged 1 to 3 months old (Control, n = 20; Inpatient (no-GA), n = 6; Inpatient GA-exposed, n = 32), and 47 were studied when they were 4 to 7 months old (Control, n = 18; Inpatient (no-GA), n = 2; Inpatient GA-exposed, n = 27). Fifty infants were studied on two occasions. Fig 1 illustrates the flow of participants through the study.

### Demographic characteristics and clinical history

The Control group were studied at a mean age of 2.3 months (SD = 0.5, n = 21), and at 4.8 months (SD = 0.4, n = 19). By definition, no participants had prior hospital admission or exposure to general anaesthesia.

The Inpatient (no-GA) group were studied at a mean age of 2.1 months (SD = 0.8, n = 7) and at 4.3 months (SD = 0.4, n = 2). All seven participants had previous hospital encounters that included an overnight stay. Of these, six were admitted to the Intensive Care Unit for

medical care. Reasons for ICU admission included feeding support, glucose monitoring, and oxygen level monitoring.

The Inpatient GA-exposed group were studied at a mean age of 2.5 months (SD = 0.9, n = 38) and at 4.8 months (SD = 0.3, n = 28). Approximately 18% of the GA-exposed sample had multiple GA exposures, ranging from 2 to 12 general anaesthetic exposures. The most common reason for multiple GA exposures was for long-gap oesophageal atresia/fistula related treatment using the Foker process [47]. At age 1 to 3 months, cumulative duration of general anaesthesia exposure ranged from 1 to 43.2 hrs (median duration = 3.8; IQR: 2 to 6.9). At age 4 to 7 months, the cumulative duration of general anaesthesia exposure ranged from 0.8 to 45.2 hrs (median duration = 5.1; IQR: 1.8 to 7.9). At age 1 to 3 months, 26 out of 38 participants (68%) had prior or ongoing admission to the Intensive Care Unit (ICU) at the time of study, ranging from 1 to 56 days. At age 4 to 7 months, 22 out of 28 participants (79%) had an ICU admission ranging from 1 to 174 days.

## Part 1: Motor responses to graded mechanical stimulus intensities

The progression of lower limb flexion responses was investigated by stimulating the plantar surface of the foot (**Fig 2**). In all infants, there was a progression of lower limb motor coordination with increasing mechanical force. **Fig 3** shows the progression of the threshold for different lower limb flexion responses to graded mechanical stimulation at 1 to 3 months, and at 4 to 7 months in the Control group. Visual inspection of the area under the curve suggested that larger motor responses (hip-flexion) were more common at lower stimulus intensities in the younger ages compared to older ages (see below for threshold statistics).

## Part 2: Number of trials to ensure reliable measurements in controls

Agreement analysis of single trial (T1) vs average of all trials (TAvg) showed good agreement for ankle-flexion threshold, and good or excellent agreement for knee and hip flexion thresholds.

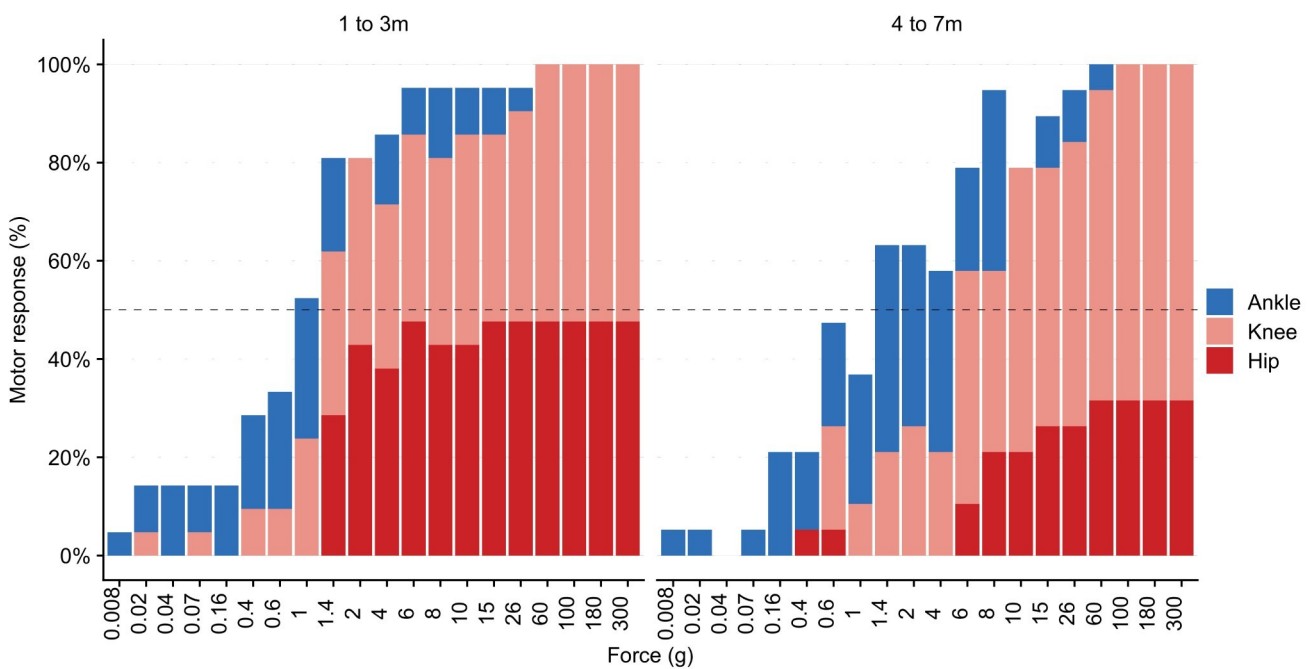

**Fig 3. Progression of the different lower limb flexion responses to graded mechanical stimulation at age 1 to 3m (n = 21), and at 4 to 7m (n = 19) in the control group.** The box plot shows the proportion of occasions that an ankle (blue), knee (pink) or hip (red) flexion was observed.

**Table 2. Agreement analysis of 'single trial' and 'average of all trial' summary measures in control subjects separated by age group.**

| Parameter | Age (m) | n | Intraclass Correlation | | | Interpretation | Paired Samples t-test | | | | | Limits of Agreement | |
|---|---|---|---|---|---|---|---|---|---|---|---|---|---|
| | | | ICC | 95% CI | | | Mean Difference | 95% CI | | t | P (adj) | | |
| | | | | Low | High | | | Low | High | | | Lower | Upper |
| Ankle | 1 to 3 | 20 | 0.79 | 0.28 | 0.92 | Good | 0.34 | 0.11 | 0.56 | 3.18 | **.03** | -1.26 | 0.59 |
| | 4 to 7 | 18 | 0.73 | 0.11 | 0.91 | Good | 0.34 | 0.11 | 0.58 | 3.12 | **.04** | -1.26 | 0.57 |
| Knee | 1 to 3 | 20 | 0.87 | 0.63 | 0.95 | Excellent | 0.16 | 0.02 | 0.29 | 2.44 | .15 | -0.72 | 0.41 |
| | 4 to 7 | 19 | 0.84 | 0.54 | 0.94 | Excellent | 0.22 | 0.02 | 0.42 | 2.30 | .20 | -1.04 | 0.60 |
| Hip | 1 to 3 | 19 | 0.80 | 0.45 | 0.92 | Good | 0.21 | -0.06 | 0.49 | 1.61 | .74 | -1.33 | 0.91 |
| | 4 to 7 | 16 | 0.98 | 0.96 | 0.99 | Excellent | 0.03 | -0.06 | 0.13 | 0.77 | 1 | -0.37 | 0.321 |

Mean difference and 95% CI data are presented as log transformed threshold values. Abbreviations: ICC = intraclass correlation coefficient; CI = Confidence Interval; P (adj) = adjusted p-value; t = t statistic.

**Table 2** presents the results of the agreement analysis. ICCs for the knee- and hip-flexion ranged from 0.80 to 0.98 and were not statistically significantly different for each age group tested. Overall, variation in the data was notably larger when examining ankle-flexion, and this was reflected by large confidence intervals and statistically significant differences. **S1 Fig in S1 File** depicts the Bland-Altman plots with Limits of Agreement boundaries. While the Limits of Agreement boundaries across all age groups contained zero for all flexion thresholds, there was a small bias towards a more conservative threshold (higher threshold) when using TAvg. These data support the utility of single trials for examination of mechanically evoked knee- and hip-flexion threshold in infants, and the use of averaging trials for ankle-flexion. We opted to implement a conservative approach to tactile sensitivity reporting and report TAvg for flexion thresholds in the remainder of this paper.

## Part 3: Tactile sensitivity and nociceptive threshold in infants, and the effects of age, prior surgery, anaesthesia or critical illness

Unlike adults, all infants responded with leg movement to mechanical stimulation of the foot, irrespective of age and cohort. **Fig 4** illustrates the relationship with increasing force and increasing magnitude of evoked activity in the Control group. Tactile sensitivity thresholds were consistently lower than nociceptive thresholds. Tactile sensitivity thresholds were associated with ankle or knee flexion across all ages. Nociceptive threshold consisting of knee or hip flexion was measurable in all infants, including the oldest at age 7-months old. **Table 3** summarises threshold data for age in the Control group.

**Fig 5** illustrates group-averaged sensory thresholds for lower limb flexion responsivity separated by age and prior hospital admission. Linear mixed model analyses were performed to assess whether infant age and prior surgery, anaesthesia or critical illness predicted sensory threshold. Tactile Sensitivity Threshold or Nociceptive Threshold were used as the dependent variable as appropriate. **Table 4** summarises the output of three models that were employed. 65 subjects with 115 study visits were included in each model (56 subjects were studied on two occasions, 9 subjects were studied on one occasion).

**Fig 6** illustrates the individual sensory thresholds, and the relationship between multiple visits.

**Model 1: Threshold ~ age + number of surgeries.** Age and number of surgeries had no significant effects on tactile sensory thresholds, which is contrary to our hypothesis. Nociceptive thresholds were significantly predicted by age at the time of study (estimate = 0.14, 95% CI = 0.06 to 0.19, t = 3.83, p < .001; **Table 4**), which is in agreement with our hypothesis.

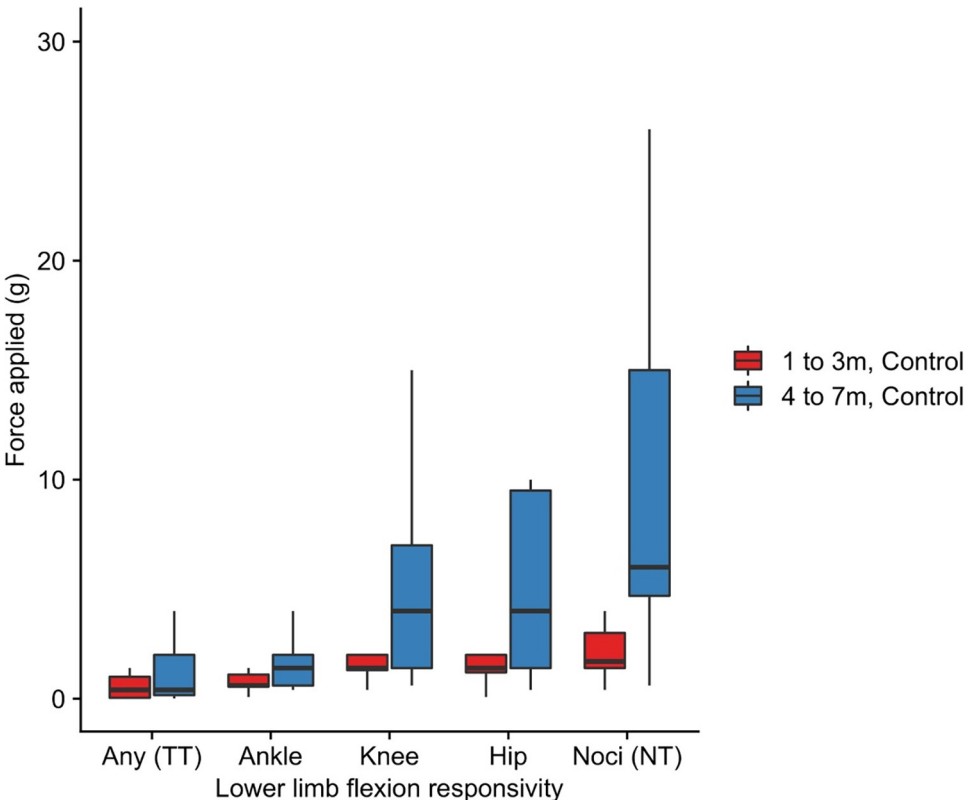

**Fig 4. Sensory thresholds for lower limb flexion responsivity when evoked by graded stimulation at age 1 to 3m (n = 21) and 4 to 7m (n = 19) in the control group.** Abbreviations: g = grams; m = months; TT = tactile threshold; NT = nociceptive threshold. Definitions: Tactile threshold = average lowest stimulus intensity required to generate any visible motion of the ankle, knee or hip; Nociceptive threshold = average stimulus intensity required to evoke brisk leg withdrawal away from the stimulus, consisting of knee- or hip-flexion.

**Model 2: Threshold ~ age + GA.** Age and duration of general anaesthesia had no significant effects on tactile sensory thresholds, which is contrary to our hypothesis (Table 4). Nociceptive thresholds were significantly predicted by age at the time of study (estimate = 0.13, 95% CI = 0.06 to 0.2, t = 3.93, p < .001; Table 4), which is in agreement with our hypothesis.

**Model 3: Threshold ~ age + GA + ICU-stay.** Age, duration of general anaesthesia, and duration of ICU stay had no significant effects on tactile sensory thresholds (Table 4).

**Table 3. Sensory thresholds (g) in the control group when tested at two ages during infancy.**

| Threshold | n | 1 to 3m Mean | SD | 95% CI Low | 95% CI High | n | 4 to 7m Mean | SD | 95% CI Low | 95% CI High |
|---|---|---|---|---|---|---|---|---|---|---|
| Tactile threshold | 21 | 0.76 | 1.29 | 0.17 | 1.34 | 19 | 1.09 | 1.26 | 0.49 | 1.70 |
| Ankle-flexion | 20 | 1.48 | 2.09 | 0.50 | 2.45 | 18 | 1.98 | 1.83 | 1.07 | 2.89 |
| Knee-flexion | 20 | 2.19 | 2.39 | 1.07 | 3.31 | 19 | 5.41 | 6.28 | 2.38 | 8.44 |
| Hip-flexion | 19 | 3.54 | 5.93 | 0.68 | 6.40 | 16 | 12.78 | 19.52 | 2.38 | 23.17 |
| Nociceptive threshold | 21 | 4.33 | 6.70 | 1.29 | 7.38 | 21 | 12.99 | 14.98 | 5.78 | 20.21 |

Threshold data are summarised from the average of all trials and provided in absolute force in grams. Abbreviations: CI = confidence interval; g = grams, m = months; n = number; SD = standard deviation. Definitions: Tactile threshold = average lowest stimulus intensity required to generate any visible motion of the ankle, knee or hip; Nociceptive threshold = average stimulus intensity required to evoke brisk leg withdrawal away from the stimulus, consisting of knee- or hip-flexion.

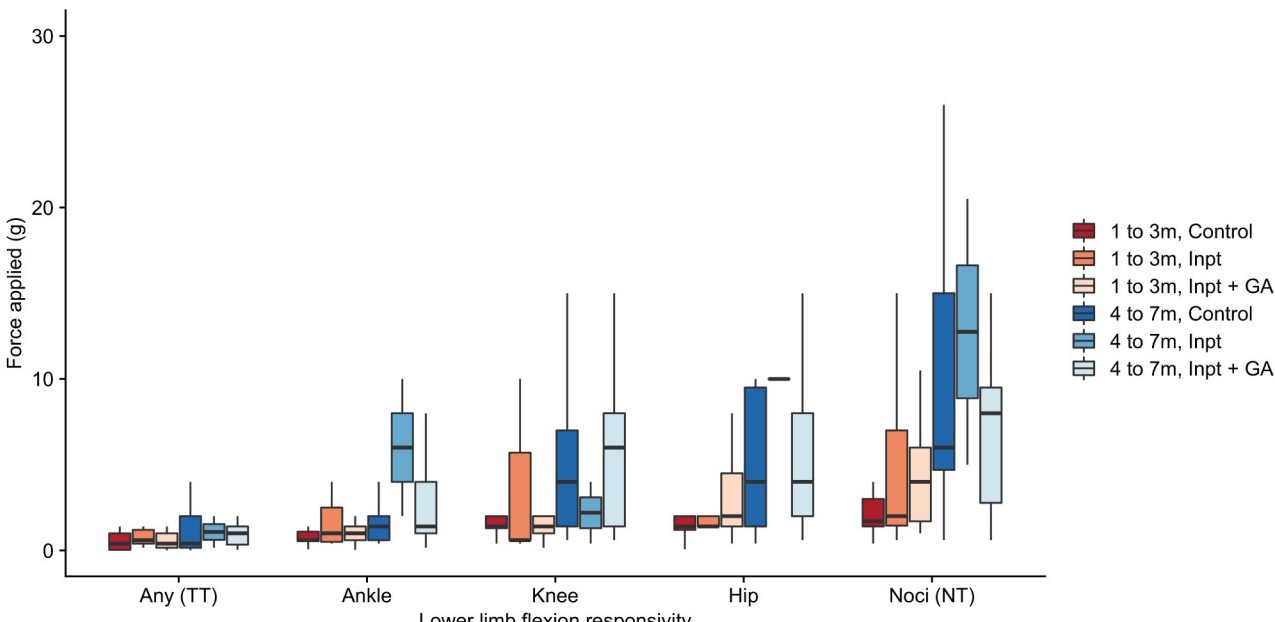

**Fig 5. Sensory thresholds for lower limb flexion responsivity when evoked by graded stimulation separated by age group and cohort.** Mixed model analysis indicated age was the sole predictor for nociceptive threshold (**Table 4**), as well as ankle, knee and hip flexion threshold in the sample (**S3 Table in S1 File**). Abbreviations: g = grams; m = months; GA = general anaesthesia exposure; TT = tactile threshold; NT = nociceptive threshold. Definitions: Tactile threshold = average lowest stimulus intensity required to generate any visible motion of the ankle, knee or hip; Nociceptive threshold = average stimulus intensity required to evoke brisk leg withdrawal away from the stimulus, consisting of knee- or hip-flexion.

Nociceptive thresholds were significantly predicted by age at the time of study (estimate = 0.13, 95% CI = 0.07 to 0.2, t = 3.99, p < .001), and not duration of GA or ICU stay, which is contrary to our hypothesis.

No statistically significant predictors of threshold were found when evaluating the effect of age, sex, and state (**S2 Table in S1 File**).

Model selection identified Model 2 as the best-fit model; this model included age and cumulative hours of general anaesthesia as fixed effects. Notably, all models had a similar performance given that AICc delta was small (< 2 AIC units). For tactile threshold modelling, Model 2 contained 37% of the cumulative model weight, and the evidence ratio (i.e. how much better Model 2 is compared to other models) for Model 1 was 1.23, and for Model 3 was 1.95. For nociceptive threshold modelling, Model 2 had 34% of the cumulative model weight, and the evidence ratio for Model 1 was 1.03 and for Model 3 was 1.6.

Model 1, 2 and 3 were also performed on ankle, knee and hip-flexion thresholds, and demonstrated age predicted threshold (**S3 Table in S1 File**).

## Discussion

In the present paper, we assessed the development of tactile processing in infants by describing in detail movements of the ankle, knee, and hip with a full range of mechanical stimuli. Furthermore, we determined the reliability of single- and averaged- threshold summary measurements and compared healthy controls to infants who have undergone hospitalisation, with and without general anaesthesia and critical illness. This work builds on previous studies that use the lower limb flexion response to evaluate tactile sensitivity and pain processing in infants beyond the newborn period [25, 27].

**Table 4. Results of linear mixed models to test effect of age and indices of prior surgery, anaesthesia or critical illness exposure on sensory thresholds.**

| Dependent variable | Model | Predictor | Fixed Effects | | | |
|---|---|---|---|---|---|---|
| | | | Estimate | 95% CI | t | P |
| Tactile sensitivity threshold | Model 1 | Intercept | -0.62 | -0.95 to -0.3 | -3.85 | < .001 |
| (N = 65, | | Age at study | 0.08 | -0.01 to -0.17 | 1.84 | .07 |
| Occasions = 115) | | Number of surgeries | -0.02 | -0.09 to 0.06 | -0.44 | .66 |
| | Model 2 | Intercept | -0.62 | -0.94 to -0.3 | -3.84 | < .001 |
| | | Age at study | 0.07 | 0 to -0.17 | 1.88 | .06 |
| | | Duration GA | -0.01 | -0.02 to 0.01 | -0.77 | .44 |
| | Model 3 | Intercept | -0.63 | -0.99 to -0.3 | -3.89 | < .001 |
| | | Age at study | 0.09 | 0 to 0.18 | 1.94 | .05 |
| | | Duration GA | -0.01 | -0.04 to 0.01 | -1.19 | .24 |
| | | Duration ICU stay | 0 | 0 to 0.01 | 0.93 | .36 |
| Nociceptive threshold | Model 1 | Intercept | 0.20 | -0.04 to 0.44 | 1.64 | .1 |
| (N = 65, | | **Age at study** | **0.13** | **0.06 to 0.19** | **3.83** | **< .001** |
| Occasions = 115) | | Number of surgeries | 0 | -0.05 to 0.06 | 0.16 | .87 |
| | Model 2 | Intercept | 0.2 | -0.04 to 0.45 | 1.69 | .09 |
| | | **Age at study** | **0.13** | **0.06 to 0.2** | **3.93** | **< .001** |
| | | Duration GA | 0 | -0.01 to 0.01 | -0.33 | .75 |
| | Model 3 | Intercept | 0.19 | -0.05 to 0.44 | 1.59 | .115 |
| | | **Age at study** | **0.13** | **0.07 to 0.2** | **3.99** | **< .001** |
| | | Duration GA | -0.01 | -0.03 to 0.01 | -0.98 | .33 |
| | | Duration ICU stay | 0 | 0 to 0.01 | 1.12 | .27 |

Random effects were low across models as indicated by ICCs ranging from 0.06 to 0.1. For tactile threshold models, $\sigma^2$ was 0.39 and $\tau_{00}$ ranged between 0.02 to 0.03. For nociceptive threshold models, $\sigma^2$ ranged between 0.21 to 0.22, and $\tau_{00}$ was 0.02.

Significant predictor (p < 0.05) is shown in bold. Abbreviations: CI = Confidence Interval; ICC = Intraclass-Correlation Coefficient; P = p value; t = t statistic; $\sigma^2$ = within-group variance; $\tau_{00}$ = between-group-variance (variation between individual intercepts and average intercept).

The study findings can be summarised as follows: (1) We show that the likelihood of ankle-, knee-, and hip-flexion all increase with greater stimulus intensity. This emphasises the importance of being explicit about which limb movements are scored by investigators studying sensory responses in infants; (2) Agreement between single trial and averaging of repeated trials when using knee- or hip- flexion (but not ankle-flexion) as key behavioural outcome measures in the infant was excellent (or borderline-excellent); (3) Nociceptive threshold increased (decreased sensitivity) as a function of age, and tactile sensitivity remained steady during the first 7 months of life; and (4) Notably, prior surgery, anaesthesia, or critical illness had a negligible relationship with either tactile or nociceptive sensory thresholds in this sample.

## Magnitude of motor response and stimulus intensity

Lower limb flexion withdrawal can be evoked in infants using standard von Frey monofilaments, unlike adults who require greater mechanical forces or direct nerve stimulation to evoke leg movement [4]. In adults, flexion withdrawal follows a modular organisation, recruiting more appropriate muscles with greater stimulus intensities to direct precise movement of the limb away from the stimulus depending on the stimulus location when elicited [48]. Neonates are capable of mounting body responses which progress in complexity in response to increasing force applied to the foot e.g. ranging from generalised body twitch to face grimacing [10]. Unlike adults, premature and term-born infants exhibit increased sensitivity and

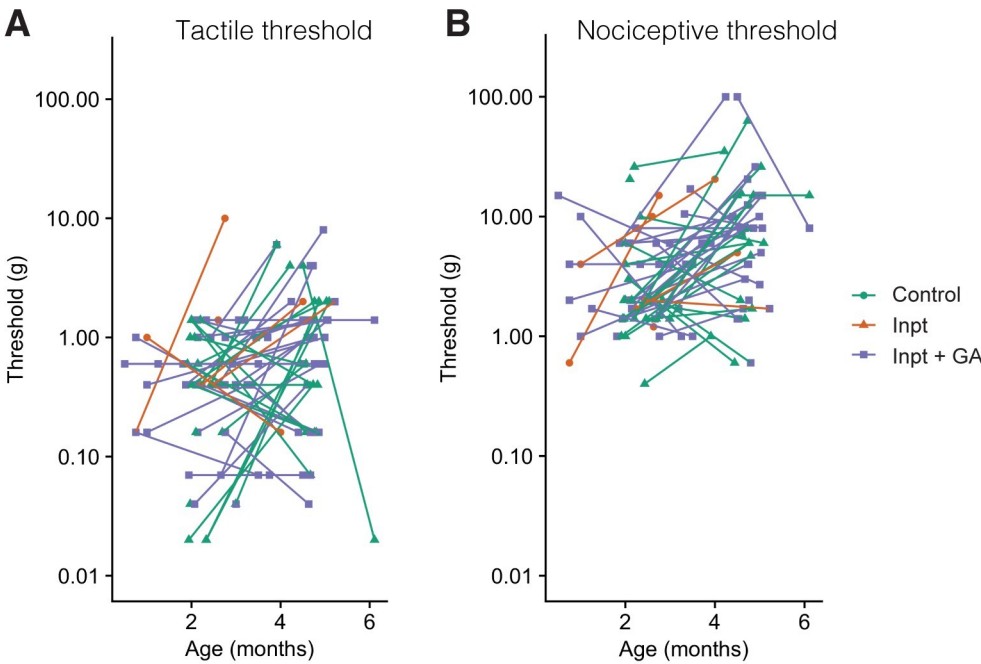

**Fig 6. Sensory thresholds measured from single and multiple study visits during infancy.** Lines indicate data taken from same subject. Colours indicate cohort. Abbreviations: g = grams; GA = general anaesthesia exposure; TT = tactile threshold; NT = nociceptive threshold. Definitions: Tactile threshold = average lowest stimulus intensity required to generate any visible motion of the ankle, knee or hip; Nociceptive threshold = average stimulus intensity required to evoke brisk leg withdrawal away from the stimulus, consisting of knee- or hip-flexion.

exaggerated motor responses to mechanical stimulation [11, 25]. These exaggerated motor responses can be elicited by innocuous and noxious stimulus intensities, and are not always nociceptive specific [11].

Gradual changes during early development likely reflect ongoing reorganisation of sensori-motor circuits. In early infancy, there is a developmental course of general movements in early infancy, with reorganization of muscle activity during spontaneous movements. Such changes include a decrease in both tonic background activity and in muscle response to passive joint movement with increasing age, which may be attributable to a reduction of the sensitivity of motor units due to spinal and supraspinal reorganisation [49, 50]. Poorly directed pain behaviour is consistent with widespread nociceptive maps in the human neonatal somatosensory cortex [51].

### Modified sensory testing protocols for paediatric populations

QST protocols are typically time-consuming to perform. Repeated trials may tax a child's patience or willingness to cooperate. Adult protocols have been modified to provide sensory assessment at the bedside for neuropathic pain [52]. We demonstrate feasibility in single trial methods to summarise tactile and nociceptive sensory threshold using knee- or hip- flexion in the infant. These brief methods can be implemented in clinical study protocols examining nerve function in pre-verbal or non-verbal patients with intact lower limb function and used to guide study design in a manner that can avoid habituation, sensitisation, or early withdrawal due to lack of subject compliance. These methods have implications for use in infants in assessing pain pathophysiology [16], behavioural disorders [17], and in clinical trials of safety and efficacy of analgesics [18, 19] and of antisense oligonucleotides for rare neurologic disorders [20].

## Cutaneous sensitivity in infancy

In our study, nociceptive threshold increases with age, consistent with previous studies in premature infants [11, 12, 25, 26], older children [5, 25, 27], and young adults [53]. Threshold of the cutaneous withdrawal reflex elicited by von Frey hairs applied to the foot is correlated with postconceptional age [12]. A longitudinal study of mechanical sensitivity in premature-born infants who were evaluated repeatedly during the first 6 weeks of life, until ~40 weeks corrected age, showed the thresholds for flexion withdrawal reflex activity increased with increasing postnatal age [26]. Beyond the neonatal period, Kuhne and colleagues show mechanical threshold continues to increase at 3, 6, and 12 months old [27].

Self-report is the preferred method of measuring sensory perception given that perception is a subjective experience. For example, typically developing verbal children and adults require relatively low stimulus intensities to report tactile sensitivity [5, 6, 20]. Standardised sensory testing studies that rely on self-report indicate mechanical sensitivity progressively matures from childhood to early adolescence (from ages 6 to 17 years) and into adulthood, as reflected by a decline in cutaneous sensitivity [4–6, 28]. Although, the magnitude of these changes in sensitivity during this later period of development are much smaller compared to the changes observed during newborn and infant periods of development. Conversely, in adults, greater stimulus intensities or invasive stimulus modalities such as skin laser, electrical, or tissue-damaging stimuli are required to evoke reflex, involuntary movement, or pain-perception [8, 54].

## Sensitivity in infants with prior surgery, anaesthesia, or critical illness

We found that prior surgery, anaesthesia, and critical illness had negligible effects on sensory threshold, which was contrary to our hypothesis. Our study sample were tested on areas that had minimal local limb damage, i.e., from heel pricks, cannulation, or surgical incision. Future work warrants testing subjects at body sites that are located closer to areas exposed to tissue damage.

## Limitations

This study has several limitations. The first limitation relates to the possibility of testing somatosensation in other parts of the body, including those adjacent to incision sites. The second limitation relates to the reliance of observational methods to assess the presence of motor activity, rather than surface electromyography as a tool to objectively and quantitatively determine the degree of muscle activity generated. Electromyography has utility in patient populations who are non-verbal with motor issues, and who may be able to perceive a tactile stimulus but may not be able to mount a reflex movement or behaviour in response. In the current study, behavioural responses were evaluated using video analysis in a frame-by-frame approach to identify subtle changes in body movement. In addition, 18% of the videos were analysed by two co-authors and good agreement was found. Future research should examine cutaneous sensitivity over a range of modalities including vibration and thermal heat/cool, and in older infants to understand the development of somatosensory processing beyond early infancy. In our experience, anecdotally we have found sensory testing using behavioural output alone to be challenging in awake children at ages 10 to 24 months due to their highly mobile and interactive nature.

## Clinical implications

The clinical implication of this work is that researchers studying infants should be explicit about the forces applied and the number of stimulus repetitions. Future studies should be

explicit in how they perform their paradigms (i.e., in scoring intensity), and to a lesser degree, their replicates, because movements vary with stimulus intensities. The study data provide normal reference ranges of sensory responses in infants that will permit early detection of tactile abnormalities in infants with neurologic diseases which influence sensory processing. These reference ranges will permit better design and interpretation of data collected from clinical trials investigating effects of hyper/hyposensitivity of topical anaesthetics and other interventions for acute pain procedures in infant populations.

## Conclusions

This study longitudinally examines tactile sensitivity and pain processing in children across the first seven months of life. We show that lower limb flexion withdrawal can be evoked in infants using standard von Frey monofilaments, unlike adults who require greater mechanical forces or direct nerve stimulation. This sensitivity is likely due to ongoing postnatal maturation of descending pathways onto the spinal cord. We demonstrate agreement in single trial methods to summarise sensory threshold that can avoid habituation, sensitisation, or early withdrawal from study protocols. This approach has potential to enhance nerve function assessment in the context of somatosensation and can be applied to infants in clinical and research settings where brief protocols are necessary. Further, we show that nociceptive threshold is associated with postnatal age during the first 7 months of life and that, in this sample at least, prior hospital experience has negligible relationship with cutaneous sensory thresholds of the feet.

## Supporting information

**S1 File. Supporting Figures and Tables provide detail on missing mechanical stimulation data (S1 Table), trial agreement analysis (S1 Fig), and additional threshold modelling analyses (S2, S3 Tables).**
(PDF)

**S1 Data.**
(XLSX)

## Acknowledgments

We would like to thank the children and their families who took part in the study. We would also like to thank Charles A. Nelson for the use of the Laboratories of Cognitive Neurosciences facilities, Carolina Donado and Adela Desowska for helpful discussions on analysis, and Siobhan Coffman for manuscript editing.

## Author Contributions

**Conceptualization:** Laura Cornelissen, Laurel J. Gabard-Durnam, Takao K. Hensch, Charles B. Berde.

**Data curation:** Laura Cornelissen, Laurel J. Gabard-Durnam, Alice Tao.

**Formal analysis:** Laura Cornelissen, Ellen Underwood, Melissa Soto, Kimberly Lobo, Charles B. Berde.

**Funding acquisition:** Laura Cornelissen, Takao K. Hensch, Charles B. Berde.

**Investigation:** Laura Cornelissen, Ellen Underwood, Laurel J. Gabard-Durnam, Alice Tao, Charles B. Berde.

**Methodology:** Laura Cornelissen, Ellen Underwood, Alice Tao, Charles B. Berde.

**Project administration:** Laura Cornelissen.

**Supervision:** Laura Cornelissen, Charles B. Berde.

**Validation:** Laura Cornelissen.

**Writing – original draft:** Laura Cornelissen, Charles B. Berde.

**Writing – review & editing:** Laurel J. Gabard-Durnam.

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
