## [Decision Letter · Decision Letter 0]

27 Sep 2022

PONE-D-22-23000Tactile sensitivity and motor coordination in infancy: effect of age, prior surgery, anaesthesia & critical illnessPLOS ONE

Dear Dr. Cornelissen,

Thank you for submitting your manuscript to PLOS ONE. After careful consideration, we feel that it has merit but does not fully meet PLOS ONE’s publication criteria as it currently stands. Therefore, we invite you to submit a revised version of the manuscript that addresses the points raised during the review process.

 The reviewers were generally positive about the study although they have some concerns requiring your attention (see their comments). 

We look forward to receiving your revised manuscript.

Kind regards,

Yury Ivanenko

Academic Editor

PLOS ONE

Journal Requirements:

Reviewers' comments:

Reviewer's Responses to Questions

**Comments to the Author**

1. Is the manuscript technically sound, and do the data support the conclusions?

Reviewer #1: Yes

Reviewer #2: Yes

Reviewer #3: Yes

Reviewer #4: Partly

2. Has the statistical analysis been performed appropriately and rigorously? 

Reviewer #1: Yes

Reviewer #2: Yes

Reviewer #3: Yes

Reviewer #4: Yes

3. Have the authors made all data underlying the findings in their manuscript fully available?

Reviewer #1: Yes

Reviewer #2: Yes

Reviewer #3: Yes

Reviewer #4: Yes

4. Is the manuscript presented in an intelligible fashion and written in standard English?

Reviewer #1: Yes

Reviewer #2: Yes

Reviewer #3: Yes

Reviewer #4: Yes

5. Review Comments to the Author

Reviewer #1: Thank you for the opportunity to review “Tactile sensitivity and motor coordination in infancy: effect of age, prior surgery, anestheisa and critical illness” This manuscript is well written and interesting and provides useful and clinically relevant. Here, the authors provide increasing force application stimuli to infants’ foot and record the motor response. They tested whether multiple testing or a single session were necessary, the effects of hospitalization and general anesthesia, and the effects of infant age. This was an impressively described study.

I rarely suggest accepting a manuscript after a first read. I certainly have some comments below, but that it overall, it was clear and worthy of publishing as is.

Introduction

Line 74: Could you add more here about this ‘critical need’? Perhaps describing more how the results of these tests are used in research or clinical purposes would help me understand why there is a critical need for better testing. You talk about this in the discussion and clinical implications, just moving it into the intro would help.

Very clear specific aims and hypotheses.

Methods

Line 145: I’m curious about the two positions- sitting vs lying down. In sitting, the flexion movement is against gravity, while lying down, it is across gravity. Does this have any affect on the LE movement?

Line 147: What ‘gross motor activity’ was recorded?

Line 149: I’m confused about what you mean by trial. Is one trial equivalent to a series of stimuli until a threshold is reached? It references S1 fig at the end of this paragraph- I like this figure and think it helps clarify the paradigm. Would you consider adding it as a figure rather than supplementary? Is there also a way on this figure to add in the Nociceptive and tactile thresholds?

Line 167: If participant is lying down, are these definitions different? Are the feet out straight during the test? And if so, how is knee flexion defined, since the thigh would likely also move?

180: Could you add an overall significance level here also?

Results

Nice tables and figures.

Discussion

Line 480- can you add some citation for this first sentence?

Line 508- I’m not sure how these last two sentences relate to your findings. They seem to contradict it. Can you expand on this?

Reviewer #2: This article aimed to study tactile sensitivity in the infant period and the associated coordination of lower limbs in children of different groups. The study was conducted on two groups of children of different ages. In addition, typically developing children were compared with infants with prior hospital experience such as surgery, anesthesia, or critical conditions.

The authors of the article described in detail the movements of the ankle, knee, and hip as responses to mechanical stimuli of varying intensity degrees. The likelihood of ankle, knee, and hip flexion has been shown to increase with greater stimulus intensity.

One of the strong points of the article is that the reliability of single and averaged summary measurements of the threshold of sensitivity was confirmed, which allows us to consider this method as sensory testing for young children.

However, it could seem that there is a low degree of objectivity in the proposed assessment, due to the fact that muscle reactions can be visually unobservable, but clearly manifested in the limb's muscle tone, in particular in children who earlier had critical conditions.

Another point for clarification is that in this article as I understood, premature babies from 32 weeks of gestational age were also included in the group of typically developing healthy children. But in the literature (Einspieler,2016; Engle WA, 2007; Natarajan G, 2016), this age is considered as a significant degree of prematurity. In this regard, I would like to know the possibility of comparing the thresholds of tactile and nociceptive sensitivity in children of two groups - premature and full-term babies.

Minor points. 1. It is possible that there is a mismatch between two values - in Tabl.1, in the second column "Control" I can read that number of participants in this group is 18, but in the line 290 it is written that n =19. 2. In the Tabl.1 In the first column it is written that "age at birth" is indicated only in months, but actually in weeks and in months.

In general, the above questions do not reduce this study's scientific and practical value. In this regard, I recommend the article for publication.

Reviewer #3: Introduction

Line #88. Inserte references in the sentence "Studies of tactile and nociceptive responses in paediatrics have focused primarily on premature and term infants, or on older children of the age and cognitive development that requires the use of verbal responses, and generally provide data from a single point in time".

Methods

It is necessary a more detailed description of the foot region of the stimulation. Was in the center of the plantar surface or in the level of the metatarsus head or in the calcaneous region or any other region?

How many researchers did the stimulation?

It is not clear how many raters carried out the motor evaluation. Were there two or more?

Please insert the confidence level of the statistics.

Reviewer #4: To the best of my knowledge, this is one of the first studies evaluating sensitivity to tactile and nociceptive stimuli in infants. The authors stimulated the foot sole of 74 infants with von Fey filament and reported the tactile/nociceptive threshold defined as the minimum stimulus (in grams) that elicited an ankle/knee/hip join flexion, or a fast leg withdrawal, respectively. The experimental question is novel and interesting, the major limitation of the study is that the methodology was rather simple and therefore it only allow the authors to record basic measurements. For example, the movement of the infants was recorded with iPhone 5S video recording. I assume that the distance of the camera and the angle with respect to the scene was highly variable across participants. Gross motor-response was assessed by observing video with a standard video player. Automatic human pose estimation using deep learning techniques (for example with open pose or deep lab cut) would be a better solution towards an objective assessment. See below for my detailed comments

Major issues

The authors uses video recordings (iPhone 5S, Apple inc., CA) to capture the subject’s motor response following each stimulus application. In future work, I would recommend to use a camera at a fixed distance to the participant and a software for automatic pose estimation. As discussed in the paragraph “Limitations”, electromyography may be also useful to detect a small response in particular in flexor muscles.

“Motor activity was inspected frame-by-frame at a 1Hz sampling rate” not clear, 1Hz was the sampling rate for the video assessment or for the recording? If (as I assume) it was the sampling rate for the assessment, which was the sampling rate for the video recording?

Model 2 and 3 are two nested models, it is redundant having both in the analysis. Compare the two models with respect to a given criterion (such as AIC or BIC) and only include the one with the better score. Probably it will be Model 2 since the third predictor Duration ICU stay is not significant and the Estimate is about zero. In Table 3 Nociceptive threshold there must be a typo, 95% of Duration ICU stay cannot range from 0.01 to 0.01 (I guess the lower CI is -0.01).

In future work it may be of interest to evaluate if similar responses can be evoked with vibration stimuli at difference amplitude and frequency (e.g. delivered with Biothesiometer or similar devices) and/or thermal stimuli.

Minor issues

Inpatient (no history of GA exposure), Inpatient (GA-exposed), and Controls (i.e., no previous hospital admission and no-GA). Report in the text of the manuscript the sample size of each group (now it is only in the figure).

No power calculation the paragraph can be removed

6. PLOS authors have the option to publish the peer review history of their article (what does this mean?). If published, this will include your full peer review and any attached files.

Reviewer #1: **Yes: **Kathryn L Havens

Reviewer #2: No

Reviewer #3: **Yes: **Givago Silva Souza

Reviewer #4: No

---

## [Author Response · Author response to Decision Letter 0]

14 Nov 2022

Please refer to the "Response-to-Reviewer" document uploaded in this manuscript revision package.

---

## [Decision Letter · Decision Letter 1]

13 Dec 2022

Tactile sensitivity and motor coordination in infancy: effect of age, prior surgery, anaesthesia & critical illness

PONE-D-22-23000R1

Dear Dr. Cornelissen,

We’re pleased to inform you that your manuscript has been judged scientifically suitable for publication and will be formally accepted for publication once it meets all outstanding technical requirements.

Kind regards,

Yury Ivanenko

Academic Editor

PLOS ONE

Additional Editor Comments (optional):

Reviewers' comments:

Reviewer's Responses to Questions

**Comments to the Author**

1. If the authors have adequately addressed your comments raised in a previous round of review and you feel that this manuscript is now acceptable for publication, you may indicate that here to bypass the “Comments to the Author” section, enter your conflict of interest statement in the “Confidential to Editor” section, and submit your "Accept" recommendation.

Reviewer #1: All comments have been addressed

Reviewer #3: All comments have been addressed

Reviewer #4: All comments have been addressed

2. Is the manuscript technically sound, and do the data support the conclusions?

Reviewer #1: Yes

Reviewer #3: Yes

Reviewer #4: Partly

3. Has the statistical analysis been performed appropriately and rigorously? 

Reviewer #1: Yes

Reviewer #3: Yes

Reviewer #4: Yes

4. Have the authors made all data underlying the findings in their manuscript fully available?

Reviewer #1: Yes

Reviewer #3: Yes

Reviewer #4: Yes

5. Is the manuscript presented in an intelligible fashion and written in standard English?

Reviewer #1: Yes

Reviewer #3: Yes

Reviewer #4: Yes

6. Review Comments to the Author

Reviewer #1: Thank you for the opportunity to review “Tactile sensitivity and motor coordination in infancy: effect of age, prior surgery, anestheisa and critical illness” again. The authors have adequately addressed all of my initial comments. I did not see any additional errors or issues, and I think the comments of other reviewers also added to the clarity and relevance of this paper. Well done!

Reviewer #3: Dear Editor, all my points were addressed, and I considered that the manuscript is suitable to be published.

Reviewer #4: The authors addressed my previous comments in particular for the analysis of the data. I recommend the publication of this manuscript.

7. PLOS authors have the option to publish the peer review history of their article (what does this mean?). If published, this will include your full peer review and any attached files.

Reviewer #1: **Yes: **Kate Havens

Reviewer #3: **Yes: **Givago da Silva Souza

Reviewer #4: No

---

## [Editor Report · Acceptance letter]

20 Dec 2022

PONE-D-22-23000R1 

Tactile sensitivity and motor coordination in infancy: effect of age, prior surgery, anaesthesia & critical illness 

Dear Dr. Cornelissen:

I'm pleased to inform you that your manuscript has been deemed suitable for publication in PLOS ONE. Congratulations! Your manuscript is now with our production department. 

Kind regards, 

on behalf of

Dr. Yury Ivanenko 

Academic Editor

PLOS ONE